# Predicted versus CT-derived total lung volume in a general population: The ImaLife study

**Hendrik J. Wisselink**[1], **Danielle J. D. Steerenberg**[1], **Mieneke Rook**[1,2], **Gert-Jan Pelgrim**[1], **Marjolein A. Heuvelmans**[3,4], **Maarten van den Berge**[5], **Geertruida H. de Bock**[4], **Rozemarijn Vliegenthart**[1,6]*

1 Department of Radiology, University Medical Center Groningen, University of Groningen, Groningen, The Netherlands, 2 Department of Radiology, Martini Hospital, Groningen, The Netherlands, 3 Department of Pulmonology, Medisch Spectrum Twente, Enschede, The Netherlands, 4 Department of Epidemiology, University Medical Center Groningen, University of Groningen, Groningen, The Netherlands, 5 Department of Pulmonology, University Medical Center Groningen, University of Groningen, Groningen, The Netherlands, 6 DataScience in Health, University Medical Center Groningen, University of Groningen, Groningen, The Netherlands

* r.vliegenthart@umcg.nl

**Data Availability Statement:** Lifelines adheres to standards for data availability. Due to ethical restrictions imposed by the Lifelines Scientific Board and the Medical Ethical Committee of the University Medical Center Groningen related to

## Abstract

Predicted lung volumes based on the Global Lung Function Initiative (GLI) model are used in pulmonary disease detection and monitoring. It is unknown how well the predicted lung volume corresponds with computed tomography (CT) derived total lung volume (TLV). The aim of this study was to compare the GLI-2021 model predictions of total lung capacity (TLC) with CT-derived TLV. 151 female and 139 male healthy participants (age 45–65 years) were consecutively selected from a Dutch general population cohort, the Imaging in Lifelines (ImaLife) cohort. In ImaLife, all participants underwent low-dose, inspiratory chest CT. TLV was measured by an automated analysis, and compared to predicted TLC based on the GLI-2021 model. Bland-Altman analysis was performed for analysis of systematic bias and range between limits of agreement. To further mimic the GLI-cohort all analyses were repeated in a subset of never-smokers (51% of the cohort). Mean±SD of TLV was 4.7 ±0.9 L in women and 6.2±1.2 L in men. TLC overestimated TLV, with systematic bias of 1.0 L in women and 1.6 L in men. Range between limits of agreement was 3.2 L for women and 4.2 L for men, indicating high variability. Performing the analysis with never-smokers yielded similar results. In conclusion, in a healthy cohort, predicted TLC substantially overestimates CT-derived TLV, with low precision and accuracy. In a clinical context where an accurate or precise lung volume is required, measurement of lung volume should be considered.

## Introduction

Pulmonary conditions are common, with two major diseases—asthma and chronic obstructive pulmonary disease (COPD)—adding up to a global prevalence of 13.1% [1]. For diagnosis and disease monitoring of COPD, several lung volumetric parameters are determined, including the total lung capacity (TLC) [2]. While the diagnosis of COPD is still based on the results of spirometry, the (separately measured) TLC is often of great importance as additional measure.

protecting patient privacy, the data are not publicly available. The data catalogue of Lifelines is publicly accessible on https://www.lifelines.nl/researcher/data-and-biobank/$6102/$6104. All international researchers can obtain metadata at the Lifelines research office (research@lifelines.nl), for which a fee is required. The Lifelines system allows access for reproducibility of the study results. For the imaging data, Lifelines or the corresponding author can be contacted to discuss options for access.

**Funding:** The position of HJW is supported by KNAW grant PSA-SA-BD-01. The current substudy is part of ImaLife. The ImaLife project is funded by an institutional research grant from Siemens Healthineers and by the Ministry of Economic Affairs and Climate (Netherlands) Policy by means of the PPP Allowance made available by the Top Sector Life Sciences & Health to stimulate public-private partnerships. The funders had no role in study design, data collection and analysis, decision to publish, or preparation of the manuscript.

**Competing interests:** The authors have declared that no competing interests exist.

There are three methods to measure the TLC. If performed at end-tidal volume, the gas dilution method (often performed with helium) and body plethysmography (often called body box), provide the functional residual capacity [2, 3] that can be added to the inspiratory capacity to obtain the TLC [4, 5]. The third method is the use of an inspiratory computed tomography (CT) scan, on which the lungs can be segmented, generally without the conducting airways [6–8]. This method relies on the assumption that the CT scan is acquired at full inspiration. Gas dilution and body box will mostly have matched results for subjects without air trapping [2]. While a CT scan allows diagnostic evaluation of both airways and parenchyma, the CT-derived total lung volume (TLV) tends to differ slightly from the first two methods, although there is a strong correlation between TLV and gas dilution or body box (r 0.87–0.90) [9–13]. Which of these three should be considered the reference standard depends on the specific clinical question or research goal [4].

To give a correct interpretation of lung volume measurements with regards to potential disease presence, severity and progress in time, expected values are required for reference [14]. Accurate prediction of TLC is of importance in some clinical applications, such as in lung transplantation where a potential lung donor is matched to a recipient [15, 16]. Recently, the Global Lung Function Initiative (GLI) published a guideline with updated models to predict the median values for several static lung volumes for healthy individuals, among which the TLC [2]. This model was endorsed by the European respiratory society (ERS) [2]. The 2021 TLC model is a generalized additive model of location shape and scale (GAMLSS), which is mathematically similar to a logistic model with age and height as parameters. It also includes a spline term that depends on sex and age. To the best of our knowledge, the GLI-2021 model has not been directly compared to CT-derived lung volume. It is unknown how well the new GLI model corresponds with CT-derived lung volumes.

The GLI models are often applied to clinical non-healthy populations, for instance to provide a baseline estimation at time of diagnosis and for follow-up purposes, expressing measurements as percentage of expected or predicted [17]. This may lead to a mismatch in clinical practice if the goal is to estimate the expected lung volume in a normal healthy person instead of the idealized reference population used by the GLI [18]. The aim of this study was to compare the outcomes of the GLI-2021 model with CT-derived lung volumes in a healthy consecutively selected sample from a Western European general population-based study cohort.

## Materials and methods

### Participant selection

CT scans in this study were acquired as part of the ongoing ImaLife study. ImaLife is embedded in Lifelines, a population-based cohort study in the northern part of the Netherlands [19–21]. Lifelines is a multi-disciplinary prospective population-based cohort study examining—in a unique three-generation design—the health and health-related behaviors of 167,729 persons living in the North of the Netherlands. It employs a broad range of investigative procedures in assessing the biomedical, socio-demographic, behavioral, physical and psychological factors which contribute to the health and disease of the general population, with a special focus on multi-morbidity and complex genetics [19]. In ImaLife, participants from Lifelines aged 45 or older are invited to undergo a low-dose chest CT scan. The institutional ethical review board from the University Medical Center Groningen gave ethical approval for the ImaLife study, and all participants provided written informed consent. The ImaLife study was registered with the Dutch Central Committee on Research Involving Human Subjects (https://www.toetsingonline.nl, NL58592.042.16). For our present study, the aim was to select a sample of 400 participants from the 1421 individuals who were scanned between June and December

2018, by consecutively including 50 women and 50 men per 5-year age group, with an age range of 45–65 years. This was done to achieve an even distribution across age. Participants with incomplete imaging data (n = 3) or missing weight information (n = 5) were replaced by continuing the sampling. Prior to the main analyses of this study, we excluded participants with COPD or self-reported lung disease (n = 110). The main analysis was performed on the 290 healthy participants (cohort H). Additional analyses were performed for the full general population sample (cohort GP, n = 400) and including only healthy never-smokers (cohort HNS, n = 147). A flowchart detailing the selection steps is shown in Fig 1.

## Lifelines parameters

This study uses data from the second assessment round of Lifelines (2014–2018), which includes questionnaire answers, as well as results from a pulmonary function test [19, 20]. The questionnaire data included smoking status, pack-years, and self-reported lung disease. The spirometric data included the Forced Expiratory Volume in 1 second ($FEV_1$) and Forced Vital Capacity (FVC), which allows determination of the GOLD stage, but this does not allow derivation of the TLC [17]. Participant height and weight were self-reported during the assessment and shortly before the CT scan, respectively. The body mass index (BMI) was computed from body weight and height. For the purposes of the analyses in this study, a participant was considered to be healthy if she/he reported no COPD, emphysema, chronic bronchitis, or asthma. In case of missing data, participants were considered ever-smokers or non-healthy, respectively.

## TLC prediction model

For this study, the ERS-endorsed GLI guideline model was used [2]. The GLI model and its predecessors were developed with the use of participants without a history of smoking or lung disease only [2, 22, 23]. The 2021 model equations look like a stratified logistic regression, although the method used to derive these equations is a generalized additive models of location shape and scale (GAMLSS) [2]. The model is based on age and height: exp(-10.1128 +0.1062*ln (age) +2.2259*ln(height) +Mspline) for women and exp(-10.5861 +0.1433*ln(age) +2.3155*ln (height) +Mspline) for men [2]. Because one of the parameters in this model (Mspline) is a variable based on sex and age, a lookup table is required to use this model, which is provided as a supplementary material to the original publication (permanently archived at http://web.archive. org/web/20210629151841/https://erj.ersjournals.com/content/erj/57/3/2000289/DC1/embed/ inline-supplementary-material-2.xlsx?download=true).

## CT scan data collection

Low-dose CT scans were acquired on a third-generation dual source CT system (Somatom Force, Siemens Healthineers) with a tube potential of 120 kV and a reference current-time product of 20 mAs (median dose-length product for cohort H 58 mGy, range 29–113) [20]. The field of view was 350 mm (with a pitch of 3.0), or, in case of a large body habitus, 400 mm (pitch 2.5). Scans were reconstructed with a slice thickness/increment of 1.0/0.7 mm, yielding approximately isotropic voxels. For this study, the reconstruction with a medium-smooth (Br40) kernel was used. The scans were acquired at inspiration according to clinical standard breath coaching.

## Image analysis

Image analysis consisted of a fully automatic extraction of the lung volume from the CT scan. This was performed with the Syngo.Via Pulmo3D package (version VB40A-HF02, Siemens

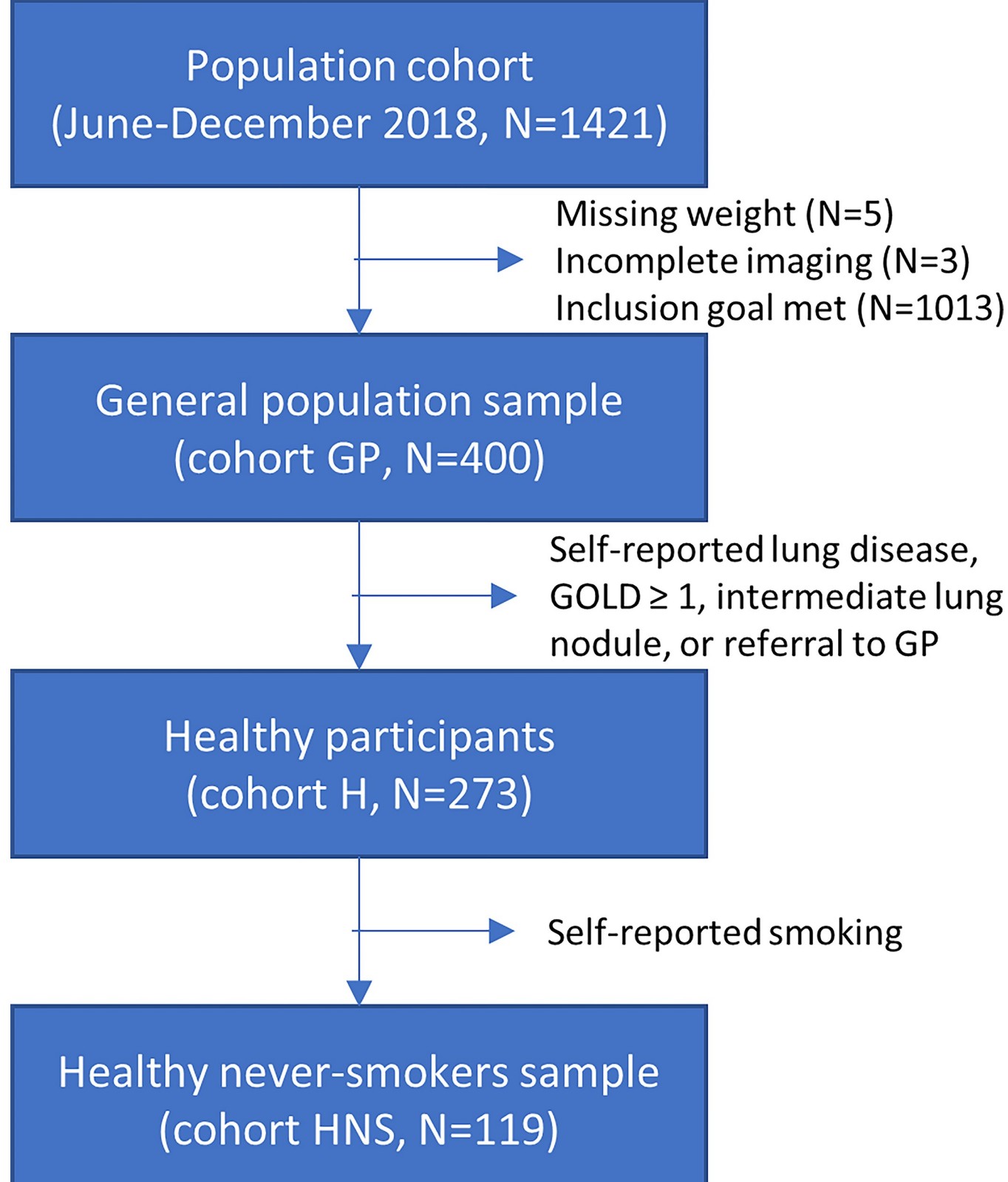

**Fig 1. Study flow chart.** Flow chart describing the selection of the GP (general population), H (healthy participants), and HNS (healthy never smokers) cohorts from the ImaLife study.

Healthineers), which did not require manual interaction. A trained researcher (DS) checked the segmentation quality. This quality check consisted of confirming all lung parenchyma was included. Lobar segmentation failures were accepted as long as the overall lung volume would be correct. An example of the segmentation result is included in the supplemental materials.

CT scans of cases with a large difference between the GLI model prediction and the CT-derived lung volume, i.e. difference being in the upper and lower 5% extremes, were visually inspected. This visual inspection was performed to ensure acquisition problems (e.g. substantial omission of an apical or caudal section of the lungs) or major pathology (e.g. severe emphysema/fibrosis) were not present and could therefore not bias the lung volume. Technical physicians (HJW and GJP, 4 years of experience in chest CT research/scan evaluation) performed visual review of these 30 cases.

## Statistical analyses

The two-sample Kolmogorov–Smirnov test was used to determine whether TLV, weight, and height for women and men are from the same distribution. Differences in age, height, weight, and TLV between women and men were tested with t-tests. Then, linear regression was used to predict TLV stratified by sex, where age, height, and weight were included as parameters. Model performance of the two models was quantified by correlating the predicted model values with the observed values of TLV by using Pearson's $\rho$ to estimate correlation and $R^2$ to estimate model fit. Then Bland-Altman analyses were performed to evaluate the systematic differences between the estimated values and the observed TLV values. The mean difference was considered as the estimated bias, and the variability is indicated by the difference between the 95% limits of agreement ($\Delta$LoA). Levene's test was used to test whether the $\Delta$LoA was the same between models and the Wilcoxon rank-sum test was used to test difference between systematic biases. All analyses were stratified by sex. The results of the Bland-Altman analysis were shown in a residual plot, showing the measured volume on one axis and the difference between TLV and TLC on the other axis.

The Bland-Altman analyses were repeated with the original consecutively selected general population sample (cohort GP, n = 400) and with only the healthy never-smokers (cohort HNS), see Fig 1. The cohort HNS was used to further mimic the cohort used for the GLI model [2].

Statistical analysis of derived data was performed with SPSS 26 (IBM). Data visualization and simple computations were done with MATLAB R2022b (Mathworks).

## Results

Visual review of cases with a large difference between predicted TLC and measured TLV did not reveal any anomalies substantial enough to warrant exclusion of the participant. Mean participant age was 54 and 53 years for women and men, respectively (Table 1). Mean weight was 74 kg for women and 86 kg for men, and mean height was 1.70 m for women and 1.84 m for men (mean BMI 25.7 kg/m² for both women and men). The prevalence of (ever) smoking was 58% for women and 55% for men (including missing data in 4 and 5 cases, respectively).

The plots in Fig 2 show TLV, height, and weight versus age. None of the scatter plots suggest a strong correlation with age.

Observed mean TLV was lower for women than for men: 4.7 L (SD 0.9 L) versus 6.2 L (SD 1.2 L), respectively (p<0.0001). Mean TLC according to the GLI-2021 was 5.7 L for women (SD 0.5 L) and 7.8 L for men (SD 0.7 L). Compared to TLV, the systematic bias of the TLC was 1.0 L for women and 1.6 L for men, indicating on average overestimation of lung volume

**Table 1. Population characteristics stratified by sex.**

| Variable | Women (N = 151) | Men (N = 139) | p value |
|---|---|---|---|
| Age (years) | 54 ± 5.5 | 53 ± 5.5 | 0.367 |
| Weight (kg) | 74 ± 12 | 86 ± 11 | <0.001 |
| Height (m) | 1.70 ± 0.07 | 1.84 ± 0.07 | <0.001 |
| Body-mass index (kg/m$^2$) | 25.7 ± 3.9 | 25.7 ± 2.9 | 0.857 |
| Smoking status | Never: 64 (42%) | Never: 63 (45%) | N.A. |
| | Past: 62 (41%) | Past: 52 (37%) | |
| | Current: 21 (14%) | Current: 19 (14%) | |
| | Missing: 4 (3%) | Missing: 5 (4%) | |
| Pack-years (current/past smokers) | 7.5 ± 7.2 | 9.2 ± 8.0 | 0.176 |
| Emphysema score (% < -950HU) | 4.1 ± 3.2 | 6.3 ± 4.1 | <0.001 |
| CT-diagnosed emphysema | None (<5%): 103 (68%) | None (<5%): 60 (43%) | N.A. |
| | Trace (5–15%): 47 (31%) | Trace (5–15%): 74 (53%) | |
| | Mild (>15%): 1 (1%) | Mild (>15%): 5 (4%) | |
| Forced expiratory volume in 1 second (FEV$_1$, L) | 2.9 ± 0.5 | 4.2 ± 0.6 | <0.001 |
| Forced vital capacity (FVC, L) | 3.8 ± 0.7 | 5.3 ± 0.8 | <0.001 |

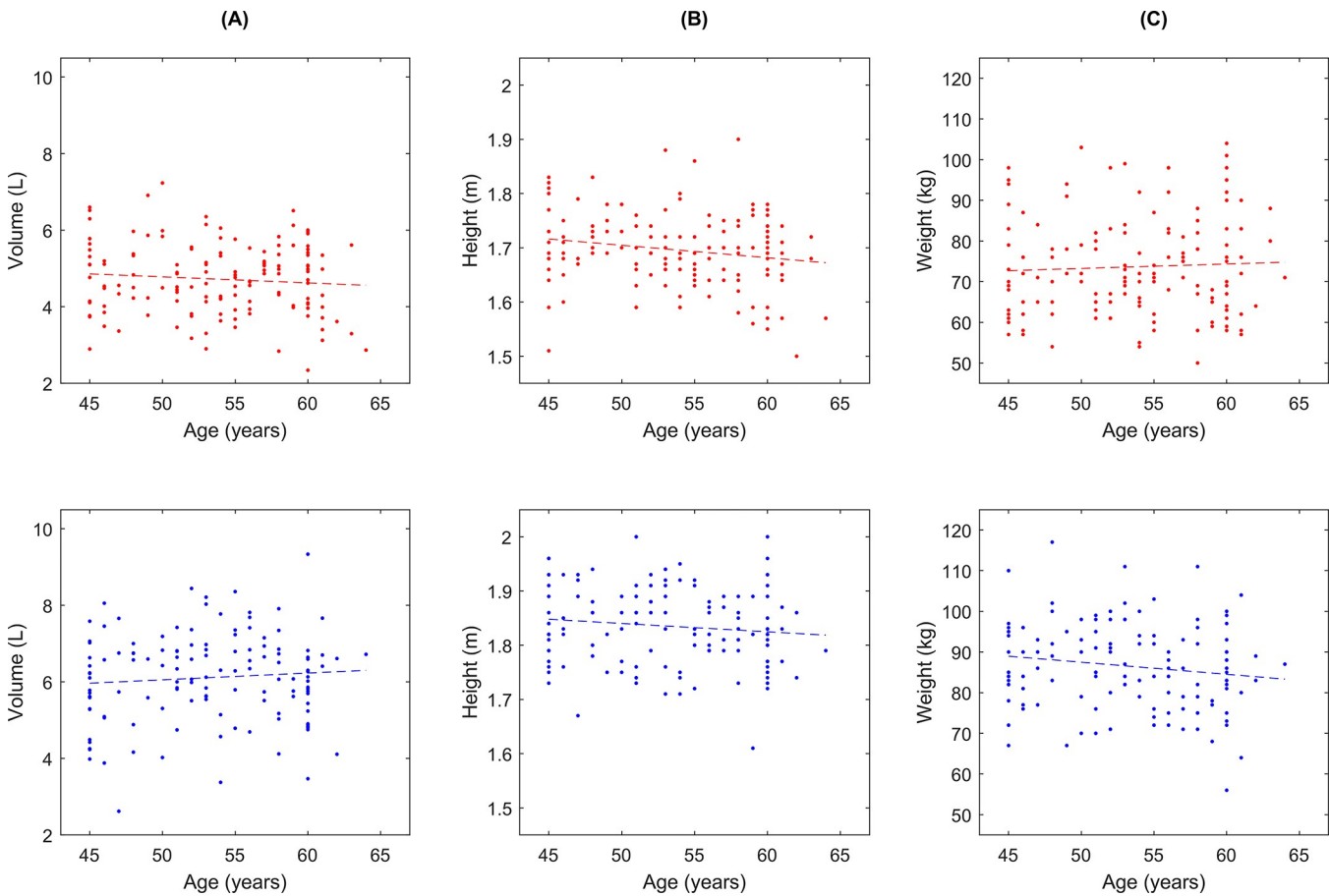

**Fig 2. Explorative scatter plots.** Explorative scatter plots showing age plotted against a) lung volume (measured on CT), b) height, and c) weight. The dotted lines are linear trend lines, determined separately for women (top row, red markers/line) and men (bottom row, blue markers/line).

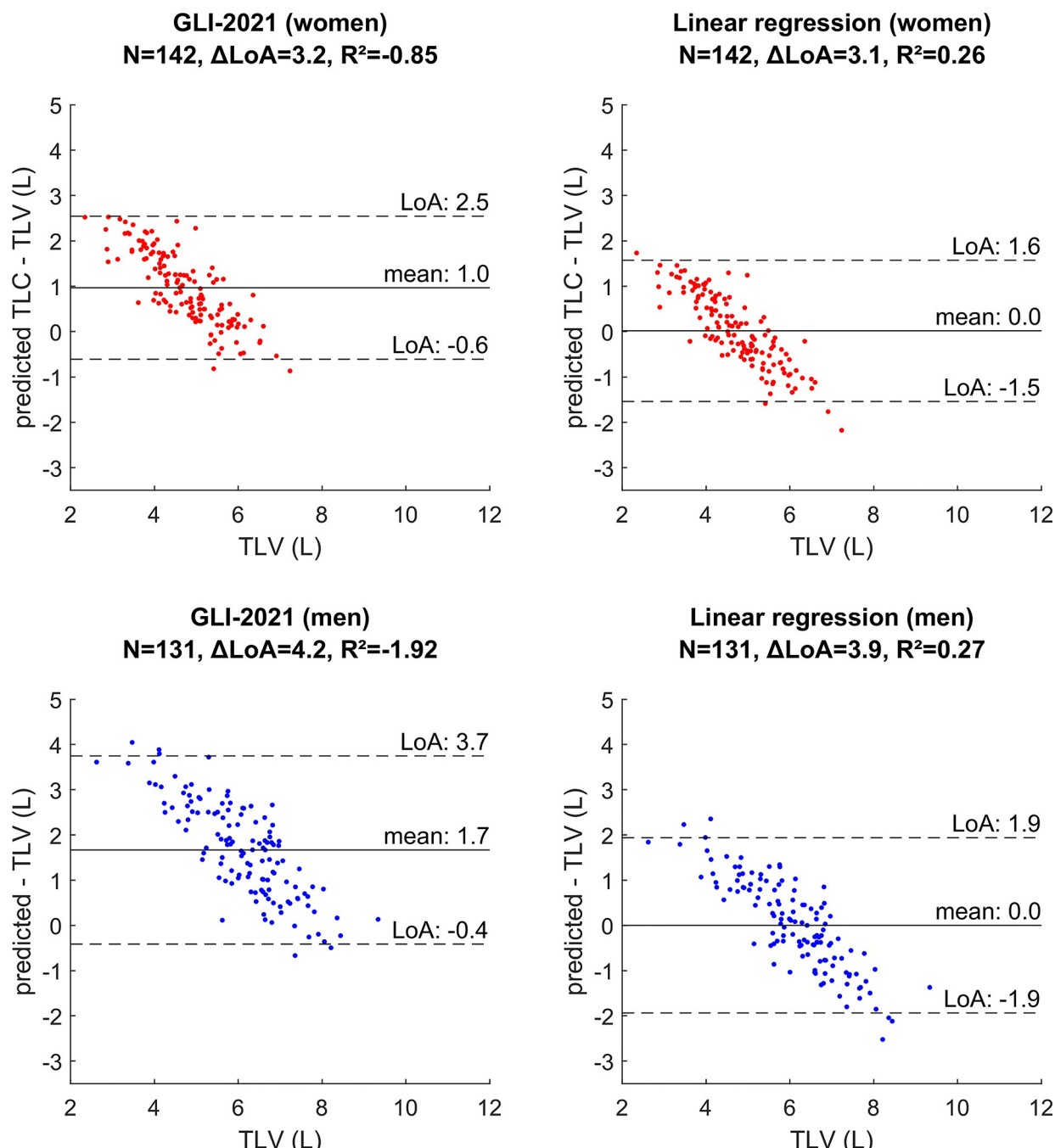

**Fig 3. Residual plots with Bland-Altman analysis stratified by sex.** Dotted lines show limits of agreement, the solid line shows the mean. GLI-2021: Global Lung Function Initiative prediction model; TLC: total lung capacity; TLV: total lung volume.

based on the GLI model. The difference ranged from 0.9 L underestimation to 4.0 L overestimation.

The residual plots in Fig 3 show the results of the Bland-Altman analysis. The difference between the limits of agreement (ΔLoA) was 3.2 L for women and 4.2 L for men, indicating large variability of GLI-model results to TLV.

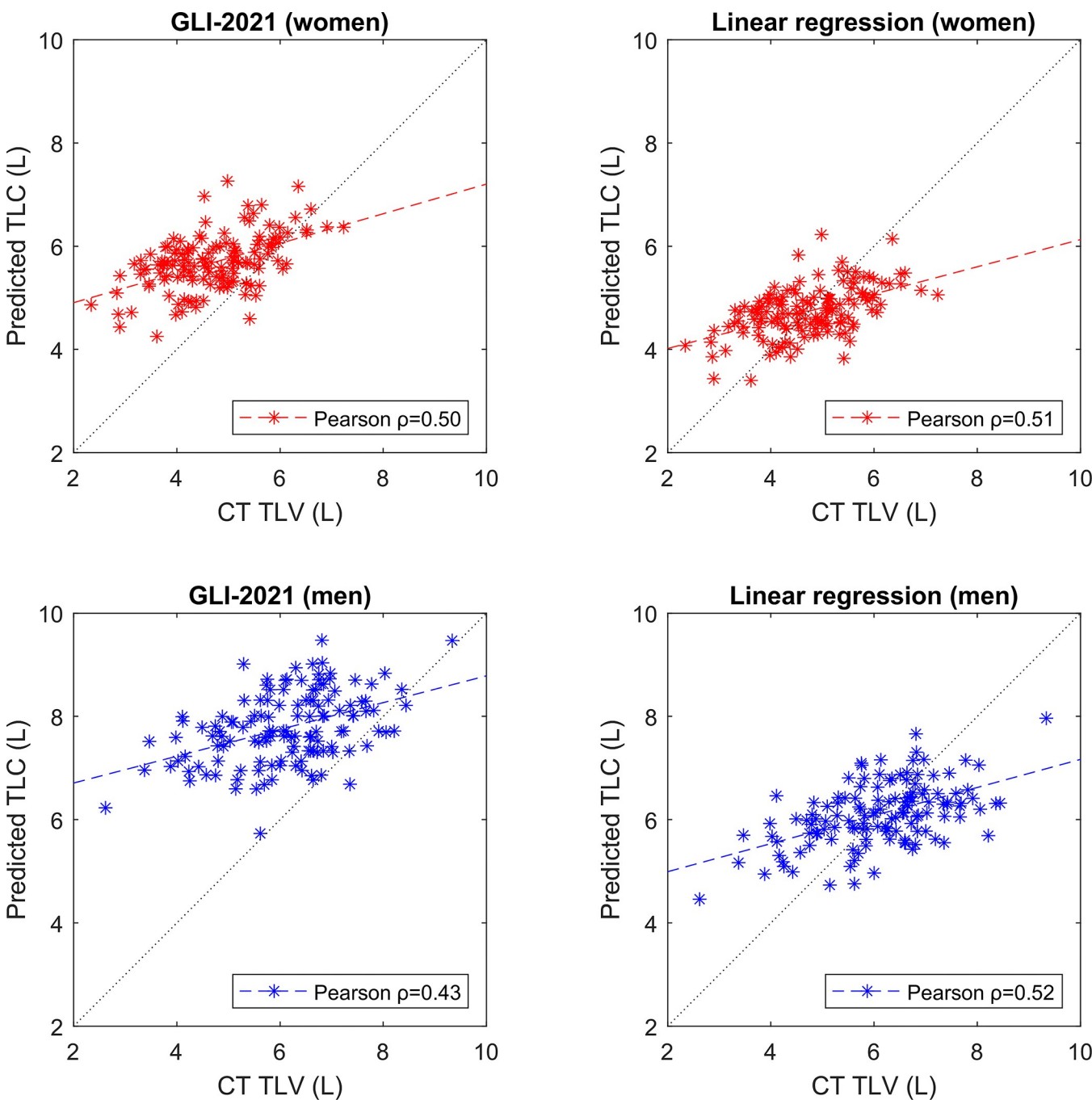

**Fig 4. Correlation plot of prediction stratified by sex.** The dotted lines are the y = x lines. Dashed lines are linear trend lines. GLI-2021: Global Lung Function Initiative prediction model; TLC: total lung capacity; TLV: total lung volume.

The correlation plots for the TLC and the TLV are shown in Fig 4. For larger lung volumes, the TLC and TLV were mostly the same, but for smaller lungs there was a progressive difference, with the predicted TLC increasingly overestimating the actual measured lung volume.

Re-including participants with lung disease or performing a sub-analysis on the healthy never-smokers did not result in significantly different systematic bias or ΔLoA (p = 0.175–0.769). A full population description of the three cohort subgroups (general population

sample, healthy participants, and healthy never-smokers) is available in S1 Table. Analysis outcomes for the three cohort subgroups including p-values are shown in S2 Table.

An optimized linear regression model based on the study population (i.e. the healthy participants) resulted in prediction formulae of lung volume for women and men (S1 Equations). The mean difference between the predictions and the TLV was -0.040 L for women and 0.026 L for men, indicating that the rounded parameters fit the data. The linear regression resulted in ΔLoA values of 3.2 L and 3.9 L, compared to the GLI-model a reduction of 1.2% (women), and 6.6% (men). ΔLoA values were not equal between the linear regression and GLI model (p>0.364).

## Discussion

This study found a substantial mismatch between the predicted total lung capacity based on the recent GLI model and CT-measured lung volume. The GLI model tended to overestimate the lung volume compared to the actual, measured TLV, by 1.0–1.6 L (24–31%), with larger overestimation in individuals with a lower TLV. The ΔLoA was high (3.2–4.2L), indicating low precision of the GLI model compared to TLV. When restricting the analyses to healthy never-smokers or expanding the analyses to include non-healthy participants, the precision and accuracy did not meaningfully change. This implies a prediction may not be sufficiently accurate or precise in clinical situations where true lung volume matters.

CT is an increasingly important modality in the evaluation of quantitative lung parameters [4]. There are suggestions that CT-derived parameters might be more sensitive than PFT measurements [24, 25]. Others have suggested that CT measurements are more reproducible than a body box [26]. This has led to the argument in a recent review by Bakker et al. that CT-derived parameters can, now or in the future replace some or all of the spirometry-based parameters [4]. The current study adds further evidence for this argument. When considering the difference between the GLI model predictions and actual CT-derived lung volume found in this study, there are several possible explanations.

Firstly, CT is normally acquired in supine position, while spirometry is performed in a sitting position; this in itself leads to a positional difference in lung volume. This is supported by the finding by Yamada et al., who compared supine and standing CT in healthy volunteers, and also reported sitting pulmonary function test measurements [9]. They reported that the mean lung volume measured in a supine position was 9.9% smaller than the mean lung volume measured in a standing position. The (unexplained) difference between standing TLV and sitting TLC was 7.5%. Since the difference between (supine) TLV and (sitting) TLC in the present study was 24–31%, this suggests only a proportion of the systematic bias may be due to the difference in position, but a third to half of the difference is likely due to overestimation by the GLI model. Furthermore, the high variability cannot be explained by positional difference.

Secondly, there are technical differences between CT and spirometry. To compare CT-derived volumes with other types of measurements, it is important to be aware of the intrinsic differences between the body box measurement (or gas dilution) and the measurement on a CT scan. Normally, the CT volume measurement will exclude conducting airways, while the volume of these airways is included for body plethysmography [8]. However, since this difference would be approximately 20 mL (trachea only) up to 60 mL (full bronchial tree), it is not clinically relevant [27, 28]. The lung segmentation might include air pockets that are not actually ventilated (or exclude air pockets that are) due to imaging artifacts. This kind of segmentation issues should be rare in the absence of pathology and was not observed in this study.

Thirdly, pathology may influence the measurements. On one hand, it may be difficult to reach maximal inspiration for patients with restrictive lung disease; on the other hand, there

may be hyperinflation in patients with COPD. Garfield et al. compared body plethysmography to CT for a cohort of COPD GOLD 3 and 4 patients [10]. They found the TLV to be 17.3% lower than the measured TLC. As we excluded patients with COPD (based on spirometry) in our study, this did not play a role in the current results.

Despite the differences outlined above, the correlation between measured TLC and TLV is high (r 0.87–0.90), regardless of the TLC measurement method (body box or gas diffusion) [9–13].

As outlined by Hall et al., the differences in predicted TLC between different models are minor in the age range 45–65. Of the six TLC prediction models spanning this age, four are within 250 mL of each other [2]. In the past decade, most prediction models (including the GLI-2012 and GLI-2021) have complex formulae, e.g. using logistic regressions with model parameters derived from splines in a separate lookup table [2, 23]. Despite this more mathematical approach, the GLI model did not result in a better fit for our study population than our linear regression. The reason is either the difference in population, or a difference in parameter choice. The cohort in this study includes participants with a smoking history and pulmonary pathology, as it is a population cohort. The linear regression includes weight as a parameter, while most other prediction models do not. Both of these differences could lead to a difference in model performance. However, we specifically performed analyses in healthy (i.e. no positive GOLD stages, no reported lung disease) and never-smoking subcohorts, to eliminate possible effects by pulmonary pathology and smoking history.

The main strength of this study is the cohort. As a sample from a population-based cohort, it matches the characteristics of the general population more closely than a hospital sample would. This is for instance important in early disease detection and monitoring and also particularly valuable in the context of lung transplantation donors where size does matter. The difference in age range between the current study population (45–65 years) and a hospital sample can be reasonably expected to be of lesser importance. This is because participants in early disease detection programs and candidates for lung donation tend to be younger than a typical hospital patient. For this study three different population types were used: a general population (cohort GP), healthy subjects (cohort H), and healthy never-smokers (cohort HNS). In general only never-smokers without pulmonary conditions (cohort HNS) have been used to develop prediction formulae [2, 22, 23]. This ignores the reality that a substantial proportion of the population is ever-smoker. Even among never-smokers there may be undiagnosed emphysema as found on CT [29]. The prediction formulae should be regarded as providing expected values for healthy never-smokers and may consequently not be accurately predicting normal values for more general populations.

The assumption that our population matches the GLI-2021 population is both a strength and a limitation of this study. The difference in age range between this study (45–65) and the GLI-2021 population (5–80) is not expected to have a large impact, as the GLI model is reasonably linear in the age range 45–65 years. To mitigate this, a visual review (including fibrosis, emphysema, and incomplete inclusion of lungs on CT) was performed on subjects with a large difference between predicted and measured lung volume. In this review no obvious disease was found. Furthermore, sub-analyses were performed with only never-smokers without pulmonary disease, as well as with the general population sample. These sub-analyses did not yield a meaningfully different variability or systematic bias.

One limitation of this study is the use of the clinical standard breath coaching, which does not completely ensure full inspiration. The breath coaching during spirometry tends to more effectively ensure maximal effort. A further limitation is the lack of external validation for our linear regression model. The same cohort was used for the creation of the model and to test the performance of the model. This limits the generalizability. Combining this limitation with

the particular height and weight of the study population, it would be interesting to repeat this study in a country with a different distribution of height and weight. Since the spirometry did not allow derivation of the TLC, it was not possible to compare a measured TLC with a measured TLV.

The current prediction models have a poor performance for lung volume as compared to actual measurements on CT in a general population cohort. Even without external validation (allowing for over-fitting of parameters), our linear regression only yielded a marginal reduction in variability of 1–7%. Combining this with the inherent population spread, as evident from the data reported by Hall et al. [2], it does not seem likely that a model with easily obtainable parameters will be able to predict lung volume with reasonable precision [18]. Future research should evaluate the possibility of machine learning to assist in accurate and precise predictions, which should be tested in populations of different ethnicity. Moreover, future research should be aimed at exploring the potential value of CT derived lung volume and other parameters for lung disease detection and monitoring.

## Conclusions and implications

This study found that there is a substantial mismatch between the GLI-predicted TLC and CT-derived TLV. The predicted TLC generally overestimates actual, measured lung volume, and has a high variability compared to TLV. A measurement (CT or otherwise) rather than a prediction should be performed in situations where size matters.

## Supporting information

**S1 Fig. Syngo.Via Pulmo 3D interface screenshot.**
(PNG)

**S1 Table. Population characteristics stratified by sex.** Participants were excluded from the HNS (healthy never-smokers) sub-cohort if they reported a smoking history, had self-reported lung disease, were determined by the spirometry to have COPD, or if any of these were missing.
(DOCX)

**S2 Table. Outcome comparison.**
(DOCX)

**S1 Equations. Results of linear regression analysis.**
(DOCX)

## Author Contributions

**Conceptualization:** Hendrik J. Wisselink, Mieneke Rook, Gert-Jan Pelgrim, Marjolein A. Heuvelmans, Maarten van den Berge, Geertruida H. de Bock, Rozemarijn Vliegenthart.

**Data curation:** Hendrik J. Wisselink, Danielle J. D. Steerenberg.

**Formal analysis:** Hendrik J. Wisselink, Danielle J. D. Steerenberg.

**Funding acquisition:** Geertruida H. de Bock, Rozemarijn Vliegenthart.

**Investigation:** Hendrik J. Wisselink, Danielle J. D. Steerenberg.

**Methodology:** Hendrik J. Wisselink, Danielle J. D. Steerenberg, Gert-Jan Pelgrim.

**Project administration:** Hendrik J. Wisselink, Mieneke Rook, Gert-Jan Pelgrim, Marjolein A. Heuvelmans, Geertruida H. de Bock, Rozemarijn Vliegenthart.

**Resources:** Maarten van den Berge, Rozemarijn Vliegenthart.

**Software:** Hendrik J. Wisselink.

**Supervision:** Hendrik J. Wisselink, Mieneke Rook, Gert-Jan Pelgrim, Rozemarijn Vliegenthart.

**Visualization:** Hendrik J. Wisselink.

**Writing – original draft:** Hendrik J. Wisselink, Danielle J. D. Steerenberg.

**Writing – review & editing:** Hendrik J. Wisselink, Mieneke Rook, Gert-Jan Pelgrim, Marjolein A. Heuvelmans, Maarten van den Berge, Geertruida H. de Bock, Rozemarijn Vliegenthart.

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
