## [Decision Letter · Decision Letter 0]

2 Jan 2023

PONE-D-22-30075Predicted versus CT-derived total lung volume in a general population: the ImaLife studyPLOS ONE

Dear Dr. Vliegenthart,

Thank you for submitting your manuscript to PLOS ONE. After careful consideration, we feel that it has merit but does not fully meet PLOS ONE’s publication criteria as it currently stands. Therefore, we invite you to submit a revised version of the manuscript that addresses the points raised during the review process.

We look forward to receiving your revised manuscript.

Kind regards,

Eman Sobh, M.D.

Academic Editor

PLOS ONE

Journal Requirements:

  "The authors received no specific funding for this work.

HJW is supported by KNAW grant PSA-SA-BD-01, RV is supported by an institutional grant from Siemens Healthineers.

Additional Editor Comments:

please respond to the reviewers comments one by one. 

Reviewers' comments:

Reviewer's Responses to Questions

**Comments to the Author**

1. Is the manuscript technically sound, and do the data support the conclusions?

Reviewer #1: Yes

Reviewer #2: Yes

2. Has the statistical analysis been performed appropriately and rigorously? 

Reviewer #1: Yes

Reviewer #2: I Don't Know

3. Have the authors made all data underlying the findings in their manuscript fully available?

Reviewer #1: Yes

Reviewer #2: Yes

4. Is the manuscript presented in an intelligible fashion and written in standard English?

Reviewer #1: Yes

Reviewer #2: Yes

5. Review Comments to the Author

Reviewer #1: Thank you for asking me review the manuscript by Wisselink and colleagues, which compared the lung total lung volume (TLC) by GLI-model with CT-derived lung volumes (TLV) in healthy participants from the northern part of the Netherlands and found that there was discrepancy between the GLI-predicted TLC and CT-derived TLV. The work itself was performed correctly in terms of statistical technique and the work is well presented. However I have read the manuscript, I am not really sure what is possible to learn that this is useful. In the present study, ages of samples were limited to 45-65 years old, which were quite different from hospital patients with lung diseases, including COPD or interstitial lung diseases. Respiratory clinicians need precise information of lung measurements in healthy older ages. Also, (supine) position and (full inspiratory) breathing might be affecting the result of lung volume measurements. I do not understand the reason of differences in position is in part due to overestimation by the GLI model. The number of non-smoking “healthy” participants was relatively small to conclude the results.

Reviewer #2: This is an interesting study looking into comparing the predicted lung volume with CT derived TLV.

I think excluding patients with COPD or lung diseases alone would not be enough, as there are other diseases that can affect lung volumes like neuromuscular diseases, liver cirrhosis with ascites, and pleural diseases that also needs to be excluded.

Relying on self-reported lung disease is questionable as many asymptomatic patients can have undetected lung diseases particularly patients with mild emphysema who can be former smoker for only 10 pack\\years. This needs to be clarified more.

The difference observed can be explained by the fact that when patients undergo lung volume assessment they are encouraged to achieve their maximal effort which as the authors stated does not happen during CT scanning and the difference in body position when utilizing these two modalities (supine vs upright), I know the authors addressed these weakness, what were the measures used to mitigate these effects on the accuracy of the study?

6. PLOS authors have the option to publish the peer review history of their article (what does this mean?). If published, this will include your full peer review and any attached files.

Reviewer #1: No

Reviewer #2: No

---

## [Author Response · Author response to Decision Letter 0]

9 Mar 2023

Editor:

Comment 3:

Thank you for stating the following financial disclosure:

"The authors received no specific funding for this work.

HJW is supported by KNAW grant PSA-SA-BD-01, RV is supported by an institutional grant from Siemens Healthineers.

Please find our clarification in the following sentences, and in the cover letter: “The position of HJW is supported by KNAW grant PSA-SA-BD-01. The current substudy is part of ImaLife. The ImaLife project is funded by an institutional research grant from Siemens Healthineers and by the Ministry of Economic Affairs and Climate (Netherlands) Policy by means of the PPP Allowance made available by the Top Sector Life Sciences & Health to stimulate public-private partnerships. The funders had no role in study design, data collection and analysis, decision to publish, or preparation of the manuscript.”

The data from the ImaLife study are part of and owned by a third-party, the Lifelines organization. Access to data can be requested by contacting the Lifelines organization (www.lifelines.nl).

 

Reviewer #1:

Thank you for asking me review the manuscript by Wisselink and colleagues, which compared the lung total lung volume (TLC) by GLI-model with CT-derived lung volumes (TLV) in healthy participants from the northern part of the Netherlands and found that there was discrepancy between the GLI-predicted TLC and CT-derived TLV. The work itself was performed correctly in terms of statistical technique and the work is well presented.

Comment 1.1:

However I have read the manuscript, I am not really sure what is possible to learn that this is useful. In the present study, ages of samples were limited to 45-65 years old, which were quite different from hospital patients with lung diseases, including COPD or interstitial lung diseases. Respiratory clinicians need precise information of lung measurements in healthy older ages.

We thank the reviewer for highlighting this important point. It is especially the requirement of precise information that motivated this study. From our population sample we selected a subpopulation of never-smokers, which we then further limited to participants who did not have any self-reported or diagnosed lung disease. Using these three different populations, we compared the CT-based results to the GLI predictions. Since the GLI predictions are based on healthy never-smokers, the results of the HNS cohort (healthy never-smokers) should have a reasonable agreement with the GLI-based predictions. The conclusion that this is not the case should serve as a warning that the presumed TLC estimate might not be as accurate as can be hoped for. The age range used in our study is shown in the comparative chart by Hall et al. to be mostly linear in all commonly used prediction models, representing a best-case scenario. While our cohort does not extend beyond 65 years old, there is no obvious reason why this warning would not extend to older ages.

Comment 1.2:

Also, (supine) position and (full inspiratory) breathing might be affecting the result of lung volume measurements. I do not understand the reason of differences in position is in part due to overestimation by the GLI model.

We thank the reviewer for highlighting this point. Since this is a clear difference between the two measurement methods, we mention this as the first possible (part) explanation for the difference between CT volume and predicted volume in the Discussion section. We also mention this in the limitations. The research performed by Yamada et al. is used as a reference for the size of this effect. Based on their research we concluded that a standing position results in approximately 10.9% higher inspiratory lung volume than a supine position. In our study we found a difference of 24-31%, meaning that a difference of 13-20% cannot be explained by the positional difference. There must either be a different explanation for this 13-20%, or this is an overestimation by the GLI model. We have added an analysis to assess the magnitude of the effect of the position on inspiratory lung volume. In the Method section we have inserted the following: “For a sensitivity analysis, the volumes reported by Yamada et al. were used to correct for the positional difference between the CT (supine) and PFT (sitting). Yamada et al. performed standing and supine CT scans on 32 healthy volunteers [9]. They found the supine and standing CT volume to be 4.3788 L and 4.8568 L, respectively. Therefore, a correction of 10.9% was added to the CT measurements in this study. This corrected TLV was then compared to the GLI-predicted TLC in a Bland-Altman analysis.”

This sensitivity analysis resulted in the addition of this paragraph to the Results: “When applying the position-correction to the CT-measured TLV, the comparison results did not meaningfully change. The mean difference was reduced slightly from 1.0/1.6 L to 0.4/0.9 L. The ΔLoA increased slightly to 3.5/4.6 L compared to the original values of 3.2/4.2 L.” In the Discussion section we have inserted: “Even after correcting for the expected difference found by Yamada et al., there is still a systematic difference and high variability”.

In the Discussion section we mention two other possible explanations for the higher lung volume in (standing) lung function testing vs (supine) chest CT: technical (segmentation-based) differences and pathology. The segmentation may cause a difference of 20-60 mL. It must be noted that this will only affect the systematic difference, as the volume of the bronchial tree and conducting airways is relatively stable. We have now added a remark to this effect in the Discussion: “It should furthermore be emphasized that this would only affect the systematic difference between CT and spirometry, and not the variability.” Lastly, the limitation section already mentioned a possible further explanatory factor, although we expect the impact to be very small: “One limitation of this study is the use of the clinical standard breath coaching, which does not completely ensure full inspiration. The breath coaching during spirometry tends to more effectively ensure maximal effort. However, in view of the healthy cohort selected for this study, and the usual ease to reach maximum inspiration in healthy individuals, we expect that the breath coaching difference will have had very limited impact on measurements.”

Comment 1.3:

The number of non-smoking “healthy” participants was relatively small to conclude the results.

We agree with the reviewer that a cohort size of 119 cases for the smallest cohort selection is relatively small, although not unusually so. Given the large limits of normality shown by Hall et al. (who used data from 6815 participants for the TLC models), it is unlikely that increasing the number of participants would substantially improve the fit results. Although the confidence in the conclusions would increase with a larger cohort size, we do not expect the conclusion itself to be different. We mitigated the potential impact of small sample in two ways. Firstly, we performed the analysis on three cohorts (from more general population to restricted healthy cohort), which led to similar results, increasing our confidence. Secondly, we avoided overly strong statements in our conclusion. 

Given the inherent spread in the population (as evident from Hall et al.), and the considerations from Guillamet (cited in manuscript, DOI: 10.1016/J.HEALUN.2022.03.012), it seems unlikely that a larger cohort would fundamentally change the outcomes. That is why in the Discussion section we advocate either for the application of a different strategy to predict lung volume, or to perform an actual measurement.

We have added a remark to the limitation section: “A final limitation concerns the cohort size. While it is unlikely the results would substantially improve with a larger cohort size, a larger number of cases would increase confidence in the conclusions, especially in the case of the HNS cohort (never-smokers without pulmonary disease).”

Reviewer #2:

This is an interesting study looking into comparing the predicted lung volume with CT derived TLV.

Comment 2.1:

I think excluding patients with COPD or lung diseases alone would not be enough, as there are other diseases that can affect lung volumes like neuromuscular diseases, liver cirrhosis with ascites, and pleural diseases that also needs to be excluded.

We thank the reviewer for highlighting this concern. While we agree with the reviewer that non-pulmonary diseases may affect lung volume, one of our goals was to establish a sample similar to the population used to develop the GLI model. Hall et al. report defining health as “never-smoked with no history of self-reported or doctor-diagnosed respiratory disease”.

To mitigate this potential limitation, outliers (i.e. cases with a large difference between predicted and measured) were visually reviewed. The description for this visual review (‘e.g. severe emphysema/fibrosis’) was not intended to be exhaustive and has now been amended to explicitly include marked pleural disease. Furthermore, in a presumed healthy population invited for (optional) CT scanning as part of an epidemiological study we do not expect severely diseased patients (such as liver cirrhosis with ascites) to have accepted the invitation. Also, one specific exclusion criterion of the ImaLife study, was that invited individuals should not have had a chest CT scan in the past year.

We have inserted a comment in the Discussion highlighting this concern to readers, which reads: “In addition to this, there are non-pulmonary conditions that may affect the lung volume, like obesity and neuromuscular diseases [Jones2006,Bach2000].”

For this revision, we additionally excluded all participants from the healthy cohort that received a follow-up scan for an intermediate lung nodule (100-300 mm³), or were referred to the GP for an incidental finding on the ImaLife chest CT scan (like significant pleural disease or aortic aneurysm). This resulted in the exclusion of an additional 17 participants. While the supplemental table showed the correct number of participants in the HNS cohort (64+63), the manuscript contained an incorrect number. This has now been rectified.

The relevant sections of the Methods now read: “Prior to the main analyses of this study, we excluded participants with COPD or self-reported lung disease, as well as participants who received a follow-up CT scan or received the advice to visit the GP due to a clinically relevant, incidentally detected finding (n=127).”

And “If a participant was invited for a follow-up scan because an intermediate size lung nodule was found or received the advice to visit the GP due to a clinically relevant, incidentally detected finding, this participant was considered non-healthy for the purpose of this study. The exact criteria for a follow-up or referral can be found in the ImaLife design paper [20].”

Comment 2.2:

Relying on self-reported lung disease is questionable as many asymptomatic patients can have undetected lung diseases particularly patients with mild emphysema who can be former smoker for only 10 pack\\years. This needs to be clarified more.

We agree with the reviewer that it is easy to miss mild disease when relying on self-reporting. That is one of the reasons why we used the PFT results in addition to self-reporting to determine the presence or absence of lung disease. To further mitigate this concern, the visual review was intended to identify a cause for a large under- or overestimation.

We have clarified this in the Method section, which now reads: “For the purposes of the analyses in this study, a participant was considered to be healthy if the spirometry did not indicate COPD (GOLD stage 0) and if she/he reported no COPD, emphysema, chronic bronchitis, asthma or interstitial lung disease.”

Comment 2.3:

The difference observed can be explained by the fact that when patients undergo lung volume assessment they are encouraged to achieve their maximal effort which as the authors stated does not happen during CT scanning and the difference in body position when utilizing these two modalities (supine vs upright), I know the authors addressed these weakness, what were the measures used to mitigate these effects on the accuracy of the study?

Please see the answer to Comment 1.2, which concerns the same issue.

---

## [Decision Letter · Decision Letter 1]

5 Jun 2023

Predicted versus CT-derived total lung volume in a general population: the ImaLife study

PONE-D-22-30075R1

Dear Dr. Vliegenthart,

We’re pleased to inform you that your manuscript has been judged scientifically suitable for publication and will be formally accepted for publication once it meets all outstanding technical requirements.

Kind regards,

Eman Sobh, M.D.

Academic Editor

PLOS ONE

Additional Editor Comments (optional):

Reviewers' comments:

Reviewer's Responses to Questions

**Comments to the Author**

1. If the authors have adequately addressed your comments raised in a previous round of review and you feel that this manuscript is now acceptable for publication, you may indicate that here to bypass the “Comments to the Author” section, enter your conflict of interest statement in the “Confidential to Editor” section, and submit your "Accept" recommendation.

Reviewer #1: All comments have been addressed

Reviewer #3: All comments have been addressed

2. Is the manuscript technically sound, and do the data support the conclusions?

Reviewer #1: Yes

Reviewer #3: Yes

3. Has the statistical analysis been performed appropriately and rigorously? 

Reviewer #1: Yes

Reviewer #3: Yes

4. Have the authors made all data underlying the findings in their manuscript fully available?

Reviewer #1: Yes

Reviewer #3: Yes

5. Is the manuscript presented in an intelligible fashion and written in standard English?

Reviewer #1: Yes

Reviewer #3: Yes

6. Review Comments to the Author

Reviewer #1: The authors have responded well according to the reviewers' comments. The manuscript was fully revised for publication to PLOS ONE.

Reviewer #3: The authors were addressed all the comments. The manuscript is a very good contribution in the prediction of lung volume based on global lung function initiative versus CT driven total lung volume.

7. PLOS authors have the option to publish the peer review history of their article (what does this mean?). If published, this will include your full peer review and any attached files.

Reviewer #1: No

Reviewer #3: **Yes: **Amr A. Abd-Elghany

---

## [Editor Report · Acceptance letter]

8 Jun 2023

PONE-D-22-30075R1 

Predicted versus CT-derived total lung volume in a general population:
the ImaLife study 

Dear Dr. Vliegenthart:

I'm pleased to inform you that your manuscript has been deemed suitable for publication in PLOS ONE. Congratulations! Your manuscript is now with our production department. 

Kind regards, 

on behalf of

Dr. Eman Sobh 

Academic Editor

PLOS ONE